# How predictability affects habituation to novelty

**Kazutaka Ueda** [1], **Takahiro Sekoguchi**[2], **Hideyoshi Yanagisawa** [2]*

1 Department of Mechanical Engineering, Creative Design Laboratory, The University of Tokyo, Tokyo, Japan, 2 Department of Mechanical Engineering, Design Engineering Laboratory, The University of Tokyo, Tokyo, Japan

☯ These authors contributed equally to this work.
* hide@mech.t.u-tokyo.ac.jp

**Data Availability Statement:** All relevant data are within the paper and its Supporting Information files.

**Funding:** This study was supported by KAKEN grant number 18H03318 and 21H03528 from the Japan Society for the Promotion of Science.

## Abstract

One becomes accustomed to repeated exposures, even for a novel event. In the present study, we investigated how predictability affects habituation to novelty by applying a mathematical model of arousal that we previously developed, and through the use of psychophysiological experiments to test the model's prediction. We formalized habituation to novelty as a decrement in Kullback-Leibler divergence from Bayesian prior to posterior (i.e., information gain) representing arousal evoked from a novel event through Bayesian update. The model predicted an interaction effect between initial uncertainty and initial prediction error (i.e., predictability) on habituation to novelty: the greater the initial uncertainty, the faster the decrease in information gain (i.e., the sooner habituation occurs). This prediction was supported by experimental results using subjective reports of surprise and event-related potential (P300) evoked by visual-auditory incongruity. Our findings suggest that in highly uncertain situations, repeated exposure to stimuli can enhance habituation to novel stimuli.

## 1 Introduction

Novelty is an essential attribute of creativity. In examining the relationship between novelty and emotion, Berlyne proposed that people feel uncomfortable when presented with either too much familiarity or too much novelty [1]. Thus, moderate novelty will make them feel comfortable. However, even if an experience is novel, one gets used to it by experiencing it repeatedly. Therefore, if one is experiencing unpleasant novel events, they should get used to them as soon as possible, whereas if one is experiencing pleasant novel events, they should be as unaccustomed as possible. Understanding one's response when repeatedly experiencing a novel event is important for maintaining an emotional response to novelty. In the field of psychology, the attenuation of a response by repeatedly experiencing a stimulus is defined as habituation [2–6]. Lécuyer (1989) postulated that the amplitude of a novelty reaction and habituation speed are linked to one's attention and speed of information processing during development [4]. Moreover, Croy et al. (2013) demonstrated that unpleasant stimuli initially caught more attention, and repeated exposure led to reduced emotional salience of unpleasant stimuli [3]. These studies indicate that habituation to novel stimuli is affected by attention.

**Competing interests:** The authors declare that the study was conducted in the absence of any commercial or financial relationships that could be construed as a potential conflict of interest.

Neuropharmacological studies using animals have investigated the neural mechanism underlying habituation to novelty [7–10]. Habituation to novelty is explained by antagonistic modulation of the excitatory nervous system (acetylcholine, adrenaline, and glutamate) and inhibitory nervous system (gamma-aminobutyric acid). As Stein's classic theory states, a novel stimulus activates the excitatory mechanism, which in turn activates the inhibitory mechanism [6]. The inhibitory mechanism becomes conditioned to the onset of the stimulus after repeated presentations; when a repeated presentation is predictable, conditioned activation of the inhibitory mechanism overrides the direct activation of the excitatory mechanism. This neuropharmacological theory shows that the predictability of novel stimuli plays an important role in the formation of habituation.

We previously developed a mathematical model of emotional dimension for novelty using information theory and a Bayesian approach [11,12]. The model formalizes arousal (primary emotional dimension) [13] as information gain obtained by experiencing events. Information gain is formalized as Kullback–Leibler (KL) divergence from prior to posterior in a Bayesian update. We found that information gain is expressed as a function of predictability (i.e., uncertainty and prediction error) when Gaussian distributions for prior and posterior are assumed. Uncertainty, an index for the familiarity of an object and the amount of knowledge an individual has, defines the variability of predictions based on prior information. Prediction error is a measure of the difference between the prior prediction and the actual sensation. Itti and Baldi (2009) proposed KL divergence from prior to posterior, which corresponds to information gain, as an index of human attention to surprising stimuli, which they demonstrated with gaze shift experiments [14]. We experimentally verified our model's prediction using subjective reports of surprise and event-related potential (ERP; parietal-dominant P300 wave). Information gain represents arousal levels because it corresponds to surprise and high-arousal state upon experiencing a novel event [12].

In this study, we assumed decrement of arousal evoked by the same event as habituation to novelty and aimed to elucidate how predictability affects habituation to novelty. We investigated the effect of predictability on habituation to novelty by applying our mathematical model of arousal [12,15,16]. We formulated habituation to novelty as a decrement in information gain (KL divergence from prior to posterior) representing arousal through prior updating by Bayesian posterior in our previous study [15,16]. In this study, we predicted the effects of the predictability (uncertainty and prediction error) on the time-course change of arousal as the primary factors constituting novelty through mathematical simulation of the model. We confirmed the biological validity of the model by experimentally demonstrating that the model prediction and the activity in the human brain were consistent. Polich et al. investigated the brain activity related to habituation to stimuli using P300, which is one component of ERP, and reported the characteristics of attenuation of brain activity by repeated stimulus presentations [17–22]. P300 (also known as P3) is a positive component that appears at a latency of approximately 250 to 600 milliseconds among the components of ERP [23]. P300 is divided into two subcomponents known as P3a and P3b. P3a reflects stimulus-driven attentional mechanisms during task processing and appears predominantly in the frontal brain regions. In contrast, P3b is involved in attention to stimuli and appears following memory processing predominantly in the parietal brain regions. In recent years, some studies have investigated habituation to stimuli using P300 [24–27], but the effects of predictability (uncertainties and prediction errors) on brain activity related to novelty habituation remain unclear. Kopp (2007) was the first to study the relationship between Bayesian inference and P300 components [28], and several empirical studies have also now been conducted [29–33]. Kolossa et al. showed the correspondence between surprise as the amount of information and P300 [32]. They reported that P3a corresponds to KL divergence from Bayesian posterior to prior (termed Bayesian

surprise), a change in the probability distribution given a new observation, particularly belief updating about hidden states. On the other hand, P3b has been reported to correspond to predictive surprise, meaning surprise about observations under the current probability distribution. In our previous study [12], we reported that information gain (KL divergence from prior to posterior) corresponded to subjectively reported surprise and parietal-dominant P300 (i.e., P3b).

In this study, we defined habituation to novelty as decrement of information gain (i.e., arousal) and aimed to elucidate the effect of predictability on them. We assumed that the change in arousal (i.e., surprise) based on the prediction made by the prior distribution, which is updated by repeated experience of novel events. Therefore, we demonstrated an experimental evidence of the model prediction using the experimental task and P3b, which has been used as an index of arousal to novel events in our previous study [12] (hereafter, P300 refers to P3b).

## 2 Modeling habituation to novelty

We mathematically formulated habituation to repeated exposure of novel stimuli based on our previously proposed model of emotional dimensions associated with novelty [12,15,16]. A novel event provides new information. We used the amount of information acquired by an event as the extent of novelty. Considering a transition before and after experiencing an event, we assumed a Bayesian update of one's belief from prior to posterior. We defined the amount of information gained from the event as KL divergence from prior to posterior, which we termed *information gain*. Information gain is a decrease in self-information averaged over posterior. In addition, the information gain also represents surprise [14] and emotional arousal [12]. When one repeatedly experiences the same event, uncertainty and surprise (i.e., information gain) to the event should decrease. Therefore, we assumed that the decrement in information gain represents habituation to a novel event.

### 2.1 Bayesian update model

Our Bayesian model assumed that one estimates a parameter $\theta$ using both one's prior $p(\theta)$ and continuous data $x \in R$ obtained by experiencing an event [12]. The Bayes' theorem updates the prior to the posterior $p(\theta|x)$ using the following equations:

$$p(\theta|x) = \frac{p(\theta)f(x|\theta)^{\alpha}}{p(x)} \propto p(\theta)f(x|\theta)^{\alpha}$$

$$p(x) = \int p(\theta)f(x|\theta)^{\alpha}d\theta = const$$

(1)

Posterior is proportional to a product of prior and a likelihood function $f(x|\theta)$ because the denominator $p(x)$, or evidence, is constant. $\alpha$ is the *learning rate* [34] that adjusts the amount of the prior update.

Assuming that one experiences the identical event and obtains the same data ($x$) $k$ times, the $k$th posterior $p_k(\theta|x)$ is proportional to a product of the initial prior and the likelihood functions when the likelihood functions are independent distributions:

$$p_k(\theta|x) = p_{k+1}(\theta) \propto p(\theta)f(x|\theta)^{\alpha k}.$$

(2)

Here, the $k$th posterior is used as $k+1$th prior $p_{k+1}(\theta)$. Assuming that one's brain encodes $n$ samples of the identical data $x$ as a Gaussian distribution $N(\mu, \sigma^2)$ with a flat prior, and using the distribution as likelihood function and the formula (2), a nonflat prior of $\mu$ following a

Gaussian distribution $N(\eta, \tau^2)$ is updated to the following Gaussian distributions:

$$p_n(\mu|x) \sim N\left(\frac{\alpha n s_{pI}\bar{x} + s_l\eta}{\alpha n s_{pI} + s_l}, \frac{s_{pI}s_l}{\alpha n s_{pI} + s_l}\right). \tag{3}$$

In this equation, $\bar{x}$ is the mean of the data, $s_{pI} = \tau^2$, and $s_l = \sigma^2$.

## 2.2 Arousal update model (habituation)

Information gain ($G_n$) in the $n$th repeated exposure of the identical continuous data or stimulus $x$ is written as Kullback–Leibler divergence from posterior to prior:

$$
\begin{aligned}
G_n &= KL(p_n(\mu|x)||p_n(\mu)) \\
&= \langle \ln p_n(\mu|x) - \ln p_n(\mu)\rangle_{p_n(\mu|x)} \\
&= \int_{-\infty}^{\infty} p_n(\mu|x) ln \frac{p_n(\mu)}{p_n(\mu|x)} d\mu
\end{aligned}
\tag{4}
$$

With the assumption of the Bayesian update in formula (2), we replace the $n$th prior by $n$-$1$th posterior.

$$G_n = \int_{-\infty}^{\infty} p_n(\mu|x)\ln\frac{p_{n-1}(\mu|x)}{p_n(\mu|x)} d\mu, \tag{5}$$

When the posterior follows the Gaussian posterior of formula (3), we derive the $n$th information gain as a function of initial parameters:

$$
\begin{aligned}
G_n &= \frac{1}{2}\left(A_n + B_n \delta_I^2\right), \\
A_n &= \frac{g_{n-1}}{g_n} - \ln\frac{g_{n-1}}{g_n} - 1, \ \ B_n = \frac{\alpha^2 s_{pI}s_l}{g_{n-1}g_n^2}, \\
g_x &= \alpha s_{pI}x + s_l.
\end{aligned}
\tag{6}
$$

We term $\delta_I = |\eta - \bar{x}|$ as the *initial prediction error* that represents the absolute difference between the prior mean and peak of the likelihood function. We term the variance of prior $s_{pI}$ as *initial uncertainty*. The variance of the data $s_l$ refers to *external noise* in the case of sensory data (i.e., stimuli). From formula (6), information gain is a function of the following three parameters: initial prediction error, initial uncertainty, and external noise.

## 2.3 Effects of initial prediction errors and initial uncertainties on the habituation to novelty

We analyzed how initial uncertainty and initial prediction error affect the decay of information gain or habituation. Fig 1 shows the decay of information gain as a function of the number of updates by repeated exposures to the same data for varied initial prediction errors when the initial uncertainty is fixed. The information gain increases with the initial prediction error at any number of updates $n \in \mathbb{N}$.

Figs 2 and 3 show the decay of information gain with the number of updates for different initial prediction errors (0.0 and 10.0). When the initial prediction error is 0.0, the larger initial uncertainties result in larger information gains. By contrast, when the initial prediction error is 10.0, larger initial uncertainties result in smaller information gain. That is, the effect of uncertainty on information gain is reversed for these two different prediction errors. In the case of n = 1 update, this reversion occurs when the relationship between different initial

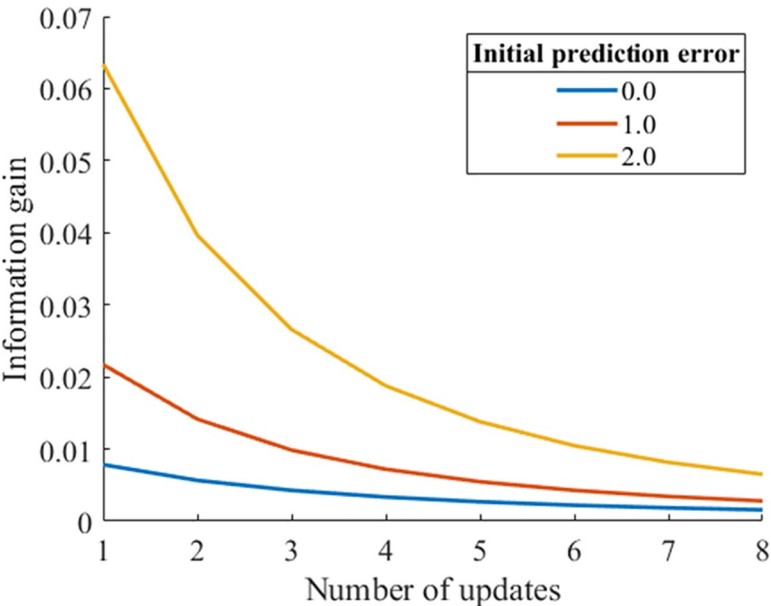

**Fig 1. Updates of information gain for different initial prediction errors (initial uncertainty = 1.0, noise = 0.5, learning rate $\alpha$ = 0.1).**

uncertainties $S_{p1}$ and $S_{p2}$ is as follows [12]:

$$s_{p1} s_{p2} > \left( \frac{S_l}{\alpha} \right)^2, \quad (s_{p1} \neq s_{p2}).$$

(7)

As shown in both Figs 2 and 3, a larger initial uncertainty decreases the information gain more significantly from $n = 1$ to $n = 2$. As shown in Fig 2, the larger information gain with a

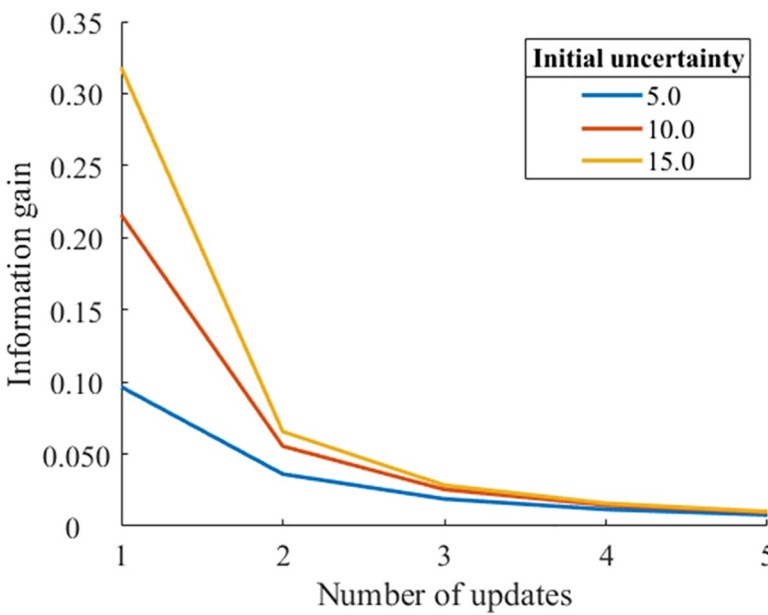

**Fig 2. Updates of information gain for different initial uncertainties (initial prediction error = 0, noise = 0.5, learning rate $\alpha$ = 0.1).**

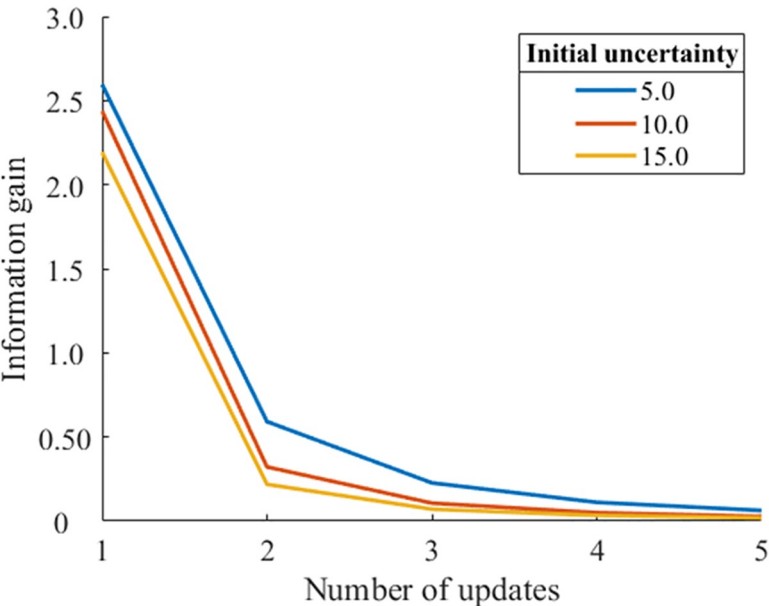

**Fig 3. Updates of information gain for different initial uncertainties (initial prediction error = 10.0, noise = 0.5, learning rate $\alpha = 0.1$).**

larger initial uncertainty quickly decreases to the same level of smaller initial uncertainty conditions. As shown in Fig 3, the information gain with a lager initial uncertainty more rapidly converges to zero. These simulation results imply that a greater initial uncertainty tends to result in a faster decay of information gain by updating, suggesting that larger initial uncertainty results in faster habituation.

Integrating information gain G with the number of updates $n$ gives the following:

$$\int G_n \mathrm{d}n = \frac{1}{2}\left(A\delta_I^{\,2} + B\right) + C$$

$$A = \frac{s_l}{s_{pI}g_n} + \frac{s_l}{\alpha s_{pI}^{\,2}} \ln \frac{g_n - \alpha s_{pI}}{g_n}$$

$$B = \left(1 - \frac{s_l}{\alpha s_{pI}}\right) \ln \frac{g_n - \alpha s_{pI}}{g_n} - n\ln \frac{g_{n-1}}{g_n}$$  (8)

$$g_n = \alpha n s_{pI} + s_l$$

C is the integration constant. Substituting infinity for $n$ of the above indefinite integral gives:

$$\lim_{n\to\infty} \int G_n \, \mathrm{d}n = \frac{1}{\alpha s_{p1}} + C$$  (9)

Eq (7) shows that the larger the value of the initial uncertainty, the smaller the sum of the information gain obtained when the stimulation is repeated indefinitely. As shown in Fig 3, the larger the value of the initial uncertainty, the smaller the initial value of the information gain. This relationship remained even when $n$ was increased to infinity. In other words, for any number of updates, the larger the value of the initial uncertainty, the smaller the information gain. Indeed, the value of the information gain when $n$ is infinite becomes smaller as the

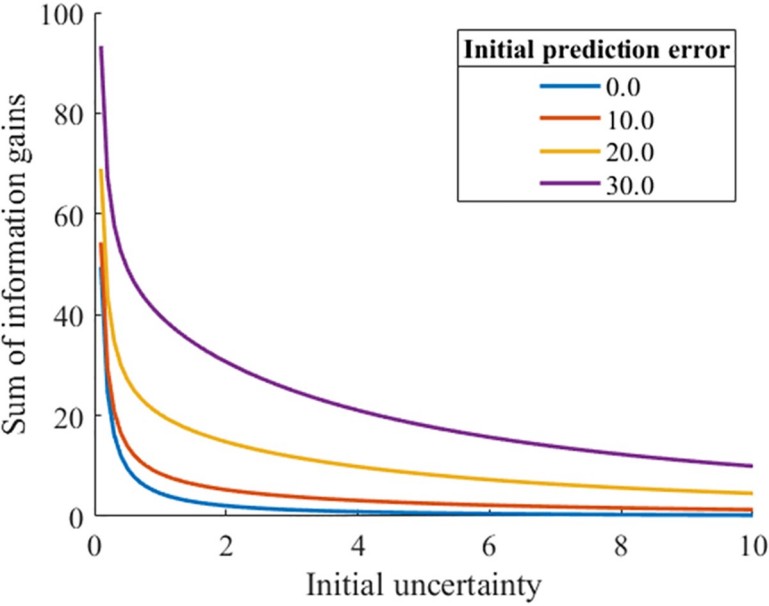

**Fig 4. Relationship between the initial uncertainty and the integrated value of the information gain.**

value of the initial uncertainty is larger, regardless of the value of the initial prediction error (Fig 4).

## 3 Experiment

We conducted an experiment using electroencephalogram (EEG) recordings and questionnaires to test our hypotheses derived from the mathematical predictions. We tested the hypothesis (H1) that the larger the initial uncertainties, the faster the surprise decays, regardless of the initial prediction error. In addition, we tested the hypothesis (H2) that larger initial uncertainties result in larger surprises when the initial prediction errors are small and smaller surprises when the initial prediction errors are large. We quantified habituation of surprise intensity using a four-level Likert scale and P300 amplitudes [12,35]. These experiments were based on the methodology in our previous study [12].

### 3.1 Materials and methods

**3.1.1 Participants.** We recruited sixteen right-handed adult males (age range: 20–27 years) who had no brain-related disorders, abnormalities associated with their eyesight, or other diseases. Handedness was assessed by the FLANDERS handedness questionnaire [36]. This study was approved by the Research Ethics Committee of the University of Tokyo, Graduate School of Engineering (KE20-59). All participants gave their consent to participate in this study.

**3.1.2 Stimuli.** We used four types of short video stimuli (duration: 2,500 ms) in which a percussion instrument was struck once and a synthesized percussive sound followed (see our previous study for details) [12]. https://www.frontiersin.org/articles/10.3389/fncom.2019. 00002/full#supplementary-material We performed an experimental manipulation of initial uncertainties due to the familiarity of percussion instruments. The clave and hand drum were used as familiar percussion instruments (i.e., low initial uncertainty, A; Table 1), and the jawbone and slit drum were used as unfamiliar percussion instruments (i.e., high initial

**Table 1. Combination of percussion instruments and percussive sounds.**

|  | Instruments | Low initial prediction error (Congruent sound, X) | High initial prediction error (Incongruent sound, Y) |
|---|---|---|---|
| Low initial uncertainty (Familiar, A) | Clave | Clave (AX) | Bell (AY) |
|  | Hand drum | Hand drum (AX) | Guiro (AY) |
| High initial uncertainty(Unfamiliar, B) | Jawbone | Jawbone (BX) | Vibraphone (BY) |
|  | Slit drum | Slit drum (BX) | Snare (BY) |

uncertainty, B; Table 1). We manipulated the initial prediction errors by the degree of congruency between the percussion instrument and the synthesized percussive sound. We used synthesized percussive sounds that were consistent with the instruments shown in congruent conditions (i.e., low initial prediction error, X; Table 1). In contrast, we used sounds that were inconsistent with the instruments in incongruent conditions (i.e., high initial prediction error, Y; Table 1). For each video stimulus, a percussion instrument was first shown in the center of the screen; 500 ms after the onset of this stimulus a percussion instrument was struck once, and a percussive sound was presented simultaneously.

**3.1.3 Procedure.** In the experiment, participants watched video stimuli while EEG recordings were taken and answered subjective feelings of surprise in an electromagnetically shielded room. The experiment consisted of 480 trials (eight videos [Table 1] × 60 presentation sets). Each of the four stimulus types (i.e., AX, AY, BX, and BY) contained two videos (Table 1). The inter-trial interval was 1,000–2,000 ms. The eight videos were presented in a random order in each set. Participants reported the intensities of their surprise using a four-level Likert scale upon listening to the percussive sounds during the first, 20th, 40th, and final presentation sets.

**3.1.4 EEG measurement.** We recorded EEGs during experimental tasks using an EEG amplifier system (eego sports, ANT Neuro) with active electrodes (sampling rate: 1000 Hz, time constant: 3 s). EEGs were recorded from electrodes positioned at the Fz, Cz, and Pz points according to the international 10–20 system [37] with reference to the nose. The impedance for all electrodes was below 60 kΩ.

**3.1.5 Analysis.** Averaged ERP waveforms were computed from 200 ms before the video stimulus onset (i.e., the start of the video) to 1,500 ms after the video stimulus onset following the application of a digital band-pass filter of 0.1–20 Hz. Waveforms were aligned to the 200 ms pre-stimulus baseline period. The averaging was performed for each participant, stimulus type (AX, AY, BX, and BY), and the number of exposure (i.e., 1–40, 41–80, and 81–120) for video stimuli. To calculate the average P300 waveforms for each participant, 40 trials were run for each exposure number. One participant's data were excluded from the ERP analysis due to excessive eye-blink artifacts. Ocular artifacts (eye movements and blinks) and muscle artifacts were removed using the Automatic Subspace Reconstruction method [38]. Any epochs containing EEG signals exceeding ± 80 μV were regarded as artifacts and were removed. P300 was defined as the largest positive peak occurring 250–600 ms after the onset of the percussive sound. The baseline-to-peak amplitude of P300 was measured at the Pz point to examine the parietal P300, which represents surprise [12,35]. To test the hypotheses, probit regression analysis and Wilcoxon's signed rank test were used to analyze the Likert scale using IBM SPSS, version 27. We used repeated-measures analysis of variance (ANOVA) to analyze P300 amplitudes. Statistical significance was defined as $p < 0.05$.

# 4 Results

Figs 5 and 6 show the average subjective scores of surprise for the number of exposure in congruent and incongruent conditions (i.e., low and high initial prediction errors), respectively.

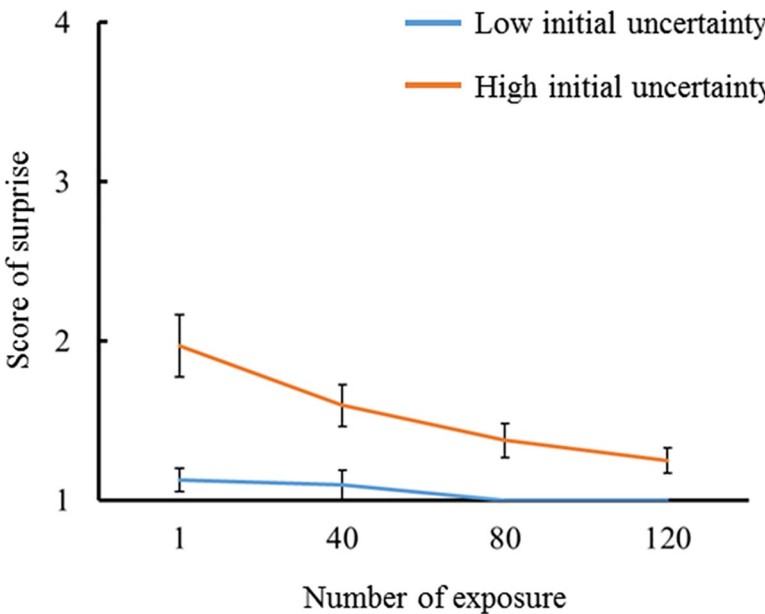

**Fig 5. Subjectively reported scores for surprise intensities in response to percussive sounds congruent with the instrument shown (i.e., low initial prediction error).** The results for familiar and unfamiliar instruments were compared at every 40 exposures (N = 16).

In both lower and higher initial uncertainties, the first exposure had the largest score, and subsequent exposures decreased the score. A probit regression analysis was conducted with the score of surprise as a dependent variable and the number of exposure as a covariate. A significant regression relationship was found under the high initial uncertainty when the initial

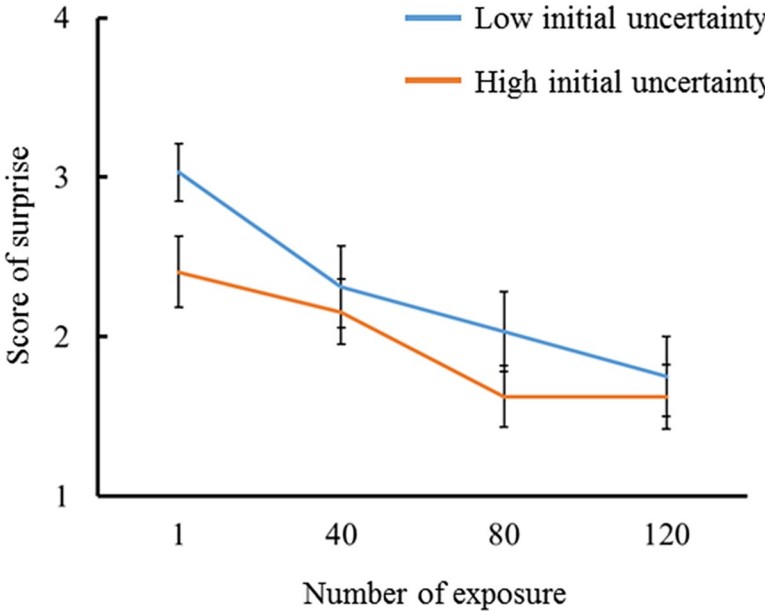

**Fig 6. Subjectively reported scores for surprise intensities in response to percussive sounds incongruent with the instrument shown (i.e., high initial prediction error).** The results for familiar and unfamiliar instruments were compared at every 40 exposures (N = 16).

prediction error was low ($p < .001$, $\beta$ = -.468, 95% confidence interval [CI] = -.212 –-.724), and under the high and low initial uncertainty when the initial prediction error was high ($p < .001$, $\beta$ = -.403, CI = -.162 –-.644; $p < .001$, $\beta$ = -.440, CI = -.199 –-.681, respectively). The surprise scores decayed with the number of exposure except for under the low initial uncertainty when the initial prediction error was low. The score of surprise decayed the fastest when the initial uncertainty was high with a low prediction error. These results partially support hypothesis H1. The score of surprise under the high initial uncertainty was greater than that under the low initial uncertainty when the initial prediction error was low. In contrast, the score of surprise under the low initial uncertainty was greater than that under the high initial uncertainty when the initial prediction error was high. For each number of exposure, the difference in the score of surprise between high and low initial uncertainty was analyzed using a Wilcoxon's signed rank test. The score of surprise was higher for the high initial uncertainty than for the low initial uncertainty for all number of exposure when the initial prediction error was low (number of exposure 1, $p = .004$, $r = .712$; number of exposure 40, $p = .008$, $r = .667$; number of exposure 80, $p = .01$, $r = .646$; number of exposure 120, $p = .011$, $r = .633$). In contrast, the score of surprise was higher for the low initial uncertainty than for the high initial uncertainty for number of exposure 1 and 80 when the initial prediction error was high ($p = .005$, $r = .707$; $p = .008$, $r = .667$, respectively). The reversal of subjective surprise for the initial uncertainty due to the initial prediction errors might reflect the simulation results, as shown in Figs 2 and 3. These results support hypothesis H2.

Fig 7 shows the grand mean ERP waveforms for the number of exposure in the four congruent and incongruent conditions (i.e., low and high initial prediction errors), respectively. In both trials with low and high initial prediction errors, the first 40 exposures had the largest P300 amplitude, and subsequent exposures attenuated the amplitude in the unfamiliar condition (i.e., high initial uncertainty). On the other hand, in the familiar condition (i.e., low initial uncertainty), the P300 amplitude gradually decreased with the increasing number of exposure in the low initial prediction error, and the P300 amplitude was maintained at the same level as that in 40 and 80 exposures and then decreased in the high initial prediction error.

Figs 8 and 9 show the average P300 amplitude for the number of exposures in congruent and incongruent conditions (i.e., low and high initial prediction errors), respectively. The P300 amplitude for the high initial uncertainty was larger than that for the low initial uncertainty in the low initial prediction error, and the P300 amplitude for the low initial uncertainty

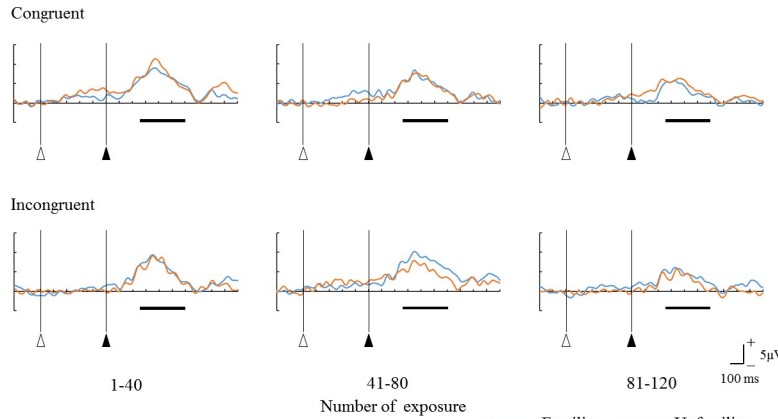

**Fig 7. Grand mean ERP waveforms for the four combinations of percussion instruments and percussive sounds at the parietal midline region (Pz).** Open triangles: The onset of film presentation. Solid triangle: The onset of beating sound presentation. The horizontal bars show the time range of 250–600 ms for the P300 latency.

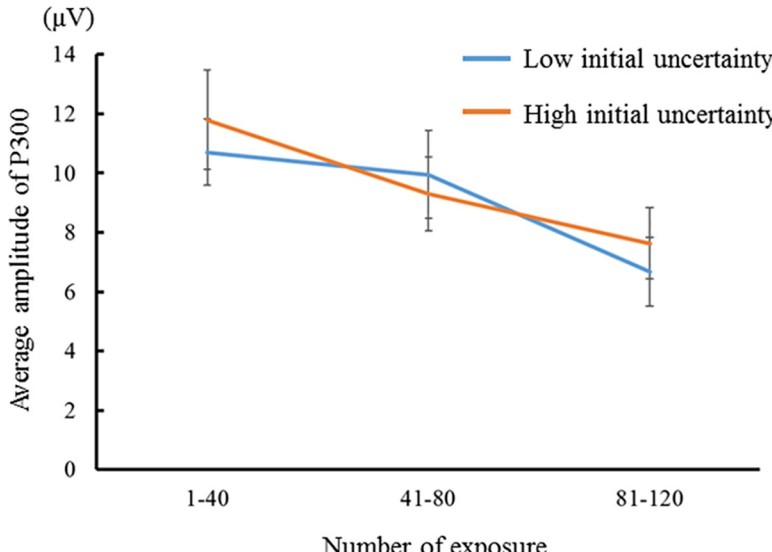

**Fig 8. P300 amplitudes evoked by percussive sounds congruent with the instrument shown (i.e., low initial prediction error).** The results for familiar and unfamiliar instruments were compared at every 40 exposures (N = 15).

was larger than that for the high initial uncertainty in the high initial prediction error. A three-factor repeated-measures ANOVA was performed with the initial uncertainty, the initial prediction error, and the number of exposures as independent variables and the P300 amplitude as a dependent variable. The main effect of the number of exposure was significant ($F(2, 28) =$ 14.600, $p < .001$, $\eta_p^2 = .510$, CI = .306 –.677). The P300 amplitude for 81–120 exposures was smaller than that for 1–40 and 41–80 exposures. The interaction effect of the initial uncertainty and the number of exposures was significant ($F(2, 28) = 6.691$, $p = .004$, $\eta_p^2 = .323$, CI = .020 –.540). The simple main effect of the number of exposures was significant for both the low

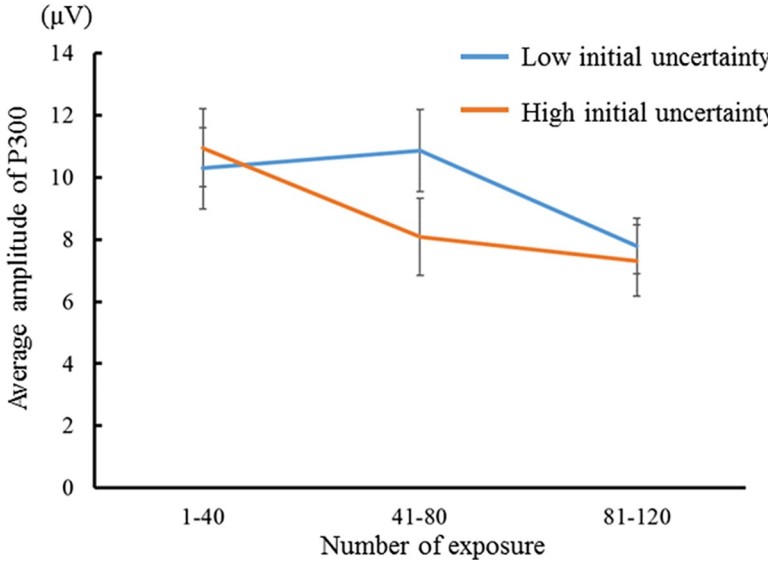

**Fig 9. P300 amplitudes evoked by percussive sounds incongruent with the instrument shown (i.e., high initial prediction error).** The results for familiar and unfamiliar instruments were compared at every 40 exposures (N = 15).

initial uncertainty ($F = 10.038$, $p < .001$, $\eta_p{}^2 = .417$, CI $= .181$ $-.606$) and the high initial uncertainty ($F = 16.686$, $p < .001$, $\eta_p{}^2 = .543$, CI $= .344$ $-.701$). The P300 amplitude for 81–120 exposures was smaller than that for 1–40 and 41–80 exposure when the initial uncertainty was low. In contrast, the P300 amplitudes for the 41–80 and 81–120 exposures were smaller than those for 1–41 exposures when the initial uncertainty was high. Therefore, the larger the initial uncertainty, the sooner the P300 decay (that is, the faster the reduction of surprise). This tendency of the 300 amplitude decay is consistent with the results of our model predictions shown in Figs 2 and 3. These results support hypothesis H1.

## 5 Discussion

In this study, we envisaged the effects of predictability on habituation to novelty by applying the Bayesian update model and assuming that a decrement in information gain (i.e., arousal) represents habituation to novelty [12,15,16]. We then conducted an experiment using subjective reports and physiological indices (P300) to provide experimental evidence for the model's predictions. The model formalized habituation as a decrement in information gain. We formalized the information gain as Kullback–Leibler divergence from prior to posterior based on Bayesian update. Based on this model, posterior is proportional to a product of prior and likelihood function. With the Gaussian prior and likelihood function, we derived the information gain as a function of three parameters: initial uncertainty, initial prediction error, and noise of sensory stimulus. Since the purpose of this study was to elucidate the effect of predictability on habituation to novelty as a decrement of information gain (decay of arousal) based on the prediction of the prior distribution as it is updated by repeated experience of novel events, we chose to examine parietal-dominant P300 (i.e., P3b), which has been used as an index of arousal to novel events [12].

We found an interaction effect between initial uncertainty and initial prediction error on habituation expressed as decrement in information gain based on mathematical simulation in the experiment using P300 and questionnaires. In particular, we found that the greater the initial uncertainty, the faster the information gain decreases and converges to zero. As previous studies have demonstrated [12,39], the initial uncertainty depends on one's prior knowledge and experience. More prior knowledge and experience results in less uncertainty. We assumed that the affinity that comes from familiarity of the object decreases the uncertainty. We conducted a P300 experiment using a set of videos of percussion instruments accompanied by synthesized percussive sounds. We manipulated the initial uncertainty with the familiarity of instruments shown and the initial prediction error with the congruency of percussive sounds. We used P300 amplitudes as an index of the participant's surprise upon hearing the percussive sounds [12]. The experimental results of P300 amplitudes support the hypothesis that the less familiar an object, the faster one becomes accustomed to novel stimuli. Consistent with the simulation results, when the uncertainty was high, the degree of information gain was greatly changed in the time transition from the initial exposure.

Brain activity related to habituation to stimuli has been investigated in many studies, including a series of studies by Polich et al. [17–22,24–27]. Our present study clarified for the first time the influence of uncertainty and prediction error on brain activity related to habituation to novel stimuli based on mathematical models. The results of this study may indicate that when attention is paid more strongly to novel stimuli (e.g., high uncertainty situation), the initial information gain increases, and accordingly, information processing is promoted, resulting in rapid habituation. In addition, Lécuyer (1989) stated that the amplitude of novelty reaction and habituation speed are linked to one's attention and speed of information processing [4], and a neuropharmacological study pointed out the relationship between predictability and

habituation of novel stimuli [6]. Our results suggest that in highly uncertain situations, repeated exposure to stimuli may increase predictability and enhance habituation to novel stimuli.

This study investigated the effects of initial uncertainty and prediction error on the habituation (decrease of arousal level) for novelty based on our mathematical models and psychophysiological experiments. We introduced the concept of Bayesian update and formulated the mechanism by defining the habituation with novelty as a decrease in information gain. The results of the simulations and experiments in this study suggest the effect of initial uncertainty on the degree of surprising attenuation for novelty and the interaction due to prediction errors. Uncertainties in this model include parameters that correspond to individual knowledge, experience, frequency of contact with events, familiarity, and typicality. Uncertainty is a factor that can explain variations in the habituation to novelty due to individual and subjective attributes. We believe it is necessary to experimentally verify the results for other parameters of uncertainty, including individuals and subjects with various attributes, in future studies.

## Supporting information

**S1 File.**
(ZIP)

**S1 Text.**
(DOCX)

## Acknowledgments

We thank Prof. Tamotsu Murakami and the members of the Design Engineering Laboratory at the University of Tokyo for supporting this project.

## Author Contributions

**Conceptualization:** Kazutaka Ueda, Hideyoshi Yanagisawa.

**Data curation:** Kazutaka Ueda, Takahiro Sekoguchi.

**Formal analysis:** Kazutaka Ueda, Takahiro Sekoguchi.

**Funding acquisition:** Hideyoshi Yanagisawa.

**Investigation:** Kazutaka Ueda, Takahiro Sekoguchi, Hideyoshi Yanagisawa.

**Methodology:** Kazutaka Ueda, Takahiro Sekoguchi, Hideyoshi Yanagisawa.

**Project administration:** Hideyoshi Yanagisawa.

**Resources:** Kazutaka Ueda, Hideyoshi Yanagisawa.

**Software:** Kazutaka Ueda, Takahiro Sekoguchi.

**Supervision:** Hideyoshi Yanagisawa.

**Validation:** Kazutaka Ueda, Hideyoshi Yanagisawa.

**Visualization:** Kazutaka Ueda, Takahiro Sekoguchi.

**Writing – original draft:** Kazutaka Ueda, Hideyoshi Yanagisawa.

**Writing – review & editing:** Kazutaka Ueda, Takahiro Sekoguchi, Hideyoshi Yanagisawa.

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
