## [Decision Letter · Decision Letter 0]

22 Sep 2020

PONE-D-20-22242

How predictability affects habituation to novelty?

PLOS ONE

Dear Dr. Yanagisawa,

Thank you for submitting your manuscript to PLOS ONE. After careful consideration, we feel that it has merit but does not fully meet PLOS ONE’s publication criteria as it currently stands. Therefore, we invite you to submit a revised version of the manuscript that addresses the points raised during the review process.

We look forward to receiving your revised manuscript.

Kind regards,

David K Sewell

Academic Editor

PLOS ONE

Journal Requirements:

2. Please modify the title to ensure that it is meeting PLOS’ guidelines (https://journals.plos.org/plosone/s/submission-guidelines#loc-title). In particular, the title should be "specific, descriptive, concise, and comprehensible to readers outside the field" and in this case it is not informative and specific about your study's scope and methodology.

Additional Editor Comments (if provided):

Dear Dr Yanagisawa,

Thank you for submitting your manuscript to PLOS ONE. I have now received reviews from two experts and have also read through the manuscript myself. You will see that the reviewers differ in their impressions of your work. Reviewer 1 identifies several major methodological and statistical issues, whereas Reviewer 2 points out several places where the writing and procedural details could be presented more clearly. In my own reading of the manuscript, I shared several of the concerns voiced by Reviewer 1, and think they are sufficiently serious to preclude publication of the work in its current form. I think additional data collection and potentially an expanded set of experimental conditions will be required to bolster confidence in the theoretical claims that you are making. I leave open the prospect of submitting a revision of the current work, noting that considerable effort will be required to successfully address the more serious issues identified by Reviewer 1.

I will summarize here what I see to be the most critical issues. The reviewers provide more detailed comments that I urge you to consider and respond to in full, should you consider revising the manuscript for resubmission.

The most pressing concern with the manuscript is the very small sample size—only 8 participants in total, with 7 producing viable data for EEG analysis. Reviewer 1 notes that the sample size is unusually small, even for an ERP study, which raises serious concerns about the study’s capacity to accurately measure the habituation effects of interest. I think the only way to really address this issue is to collect additional data, ideally replicating the original study but with much larger sample.

Reviewer 1 also raises concerns about the analysis of the surprise rating data—there is a clear floor effect for the congruent stimuli—which is very likely to be contributing to the key interaction effect described on pg. 11. The reviewer notes that adopting a different method of analysis might be a more appropriate way of analyzing the data, and I encourage you to pursue this. I also wonder whether it might be helpful to examine an “intermediate” level of congruency in an attempt to avoid floor effects on surprise. For example, could you present a modified sound produced by an instrument such that the instrument could not naturally that sound. (Perhaps applying something like a gated reverberation effect could create such an intermediate level of congruency, though feel free to tell me if this is off base.)

A further concern with the data analysis lies with the binning. I found it a little unusual that the bins were so coarse-grained, given your interest in what appears to be a fairly rapid-onset habituation effect. A better strategy for tackling your key research question would seem to be to opt for more fine-grained aggregation to detect effects that arise early in the learning process. This issue is potentially entangled with the small sample size problem. Reviewer 1 suggests single-trial regression as a potentially more appropriate method of analysis for the kind of data you have.

A final concern I had—also shared by Reviewer 1—is theoretical in nature. You provide a detailed quantitative overview of your Bayesian model, yet you investigate only qualitative predictions (e.g., an interaction effect on P300 amplitude). Finding these effects in the data does not selectively support the information gain hypothesis, and reliance on this kind of reverse inference is not justified. At minimum, there needs to be some consideration of alternative accounts for the data.

Should you choose to revise and resubmit your work, I will send the manuscript back out for review, as there is a clear need for additional data collection.

Yours sincerely,

David Sewell

Reviewers' comments:

Reviewer's Responses to Questions

**Comments to the Author**

1. Is the manuscript technically sound, and do the data support the conclusions?

Reviewer #1: Partly

Reviewer #2: Yes

2. Has the statistical analysis been performed appropriately and rigorously? 

Reviewer #1: No

Reviewer #2: Yes

3. Have the authors made all data underlying the findings in their manuscript fully available?

Reviewer #1: Yes

Reviewer #2: Yes

4. Is the manuscript presented in an intelligible fashion and written in standard English?

Reviewer #1: No

Reviewer #2: Yes

5. Review Comments to the Author

Reviewer #1: Thank you for the opportunity to review this interesting manuscript by Ueda and colleagues. The manuscript reports an event-related potential study of habituation to novelty, using a format that combines both computational simulation and empirical work. The primary finding of the manuscript is of an interaction effect between participants' initial uncertainty regarding a sensory stimulus and the rapidity with which habituation occurs. This interaction is predicted by a Bayesian updating model, and the manuscript reports two pieces of evidence in support of this prediction: one involving self-reports of surprise, and one involving the amplitude of the P300 component of the event-related potential.

In general, I thought that the general approach of the manuscript was sound, and the predictions from simulations were reasonable and sensible. However, I have several major concerns regarding the empirical approach of this manuscript. Of these, my feeling is that all but one might be addressed in a revision. Unfortunately, my feeling is that the remaining point, which concerns sample size and statistical power (point #1 below) severely undermines the interpretability of the empirical results reported in this manuscript.

## Critical point

1. The empirical section of this manuscript reports a sample of N=8 participants. After exclusion of 1 participant due to excessive eye-blink artefacts, therefore, the final results of this manuscript are derived from a sample of N=7 participants. The effect of this is that the manuscript is severely underpowered to estimate the effect size predicted by its simulations (between initial uncertainty and trial number). This problem is not unique to this manuscript; for instance, Button et al. (Nature Reviews Neuroscience, 2013) discuss the prevalence of underpowered studies in neuroscience more broadly. There are many consequences of underpowered studies, but the most important one for the present study is that significant results deriving from small samples are likely to significantly overestimate true effect sizes in the population. Notably, this study is underpowered even by the modest standards of ERP research; for instance, Clayson et al. (Psychophysiology, 2019) report the median sample size for ERP studies as 21 participants (3 times that of the sample in this manuscript). Unfortunately, short of replicating results in a larger sample, I believe that this represents a critical shortcoming of the manuscript that precludes interpretation of results.

## Major points

2. The results of the manuscript are presented as a test of a specific computational model of the P300 component of the ERP (in which information gain is operationalised as Kullback-Leibler divergence between posterior and prior beliefs). However, the test of this model presented in the manuscript is qualitative, rather than quantitative: in both the simulations and the empirical data, there is an interaction between trial number and initial uncertainty. However, such an interaction might also result from other cognitive mechanisms if we assume that their effect is to cause a roughly multiplicative decay in P300 amplitude over time. To give one example, an effect of participant fatigue might also produce the interaction between trial number and initial uncertainty reported in the manuscript.

A far stronger test of the model's predictions would be, rather than discretising trials into 40-trial bins, to run a single-trial regression of P300 amplitudes onto information gain as measured by the manuscript's metric. If this model outperforms one in which P300 amplitude is predicted as a function of trial number (rather than KL divergence), this would constitute stronger evidence for the manuscript's conclusion. This would also be consistent with other recent information-theoretic and Bayesian approaches to analysis of the P300 component (see point 4 below).

3. An interaction between trial number and initial uncertainty is reported for participants' self-reported surprise ratings. However, I believe that this may be a statistical artefact resulting from the fact that self-report ratings of congruent items are at a floor level (see Figure 5). Indeed, from inspection of the self-report scores in the accompanying supplemental material, it appears that the overwhelming majority of participants reported surprise levels of '1' (no surprise) for these conditions. In this case it is not surprising that an interaction would occur with initial uncertainty, since surprise can hardly be expected to decrease any further for instruments high in initial certainty. I would also note that in this case, the assumption of the ANOVA (normality of residuals) is violated, and the manuscript might be better served employing a different analytic approach (e.g., probit regression; see Liddell & Kruschke, 2018, Journal of Experimental Social Psychology).

4. The manuscript's Bayesian model of the amplitude of the P300 is an interesting one, but it does not engage with a body of prior research on Bayesian models of the P300 component of the event-related potential. A Bayesian model of the P300 was first proposed by Kopp (2008; In M. K. Sun (Ed.), Cognitive sciences at the leading edge (pp. 87–96)), and a number of empirical papers testing this and other Bayesian P300 models have been published in the last 5 years (e.g., Kolossa et al., 2013, Frontiers in Human Neuroscience; Kolossa, Kopp & Fingscheidt, 2015, NeuroImage; Bennett et al., 2016, eNeuro; Kopp et al., 2016, Cognitive, Affective, and Behavioural Neuroscience; Bennett et al., 2019, Psychophysiology). The manuscript would be strengthened by engaging with these prior papers that prefigure the ideas proposed here.

## Minor points

- Page 4, lines 86-87: "Accordingly, the information gain represents a decrease in uncertainty by experiencing an event". This is not quite correct; KL divergence measures the similarity between two probability distributions. These distributions may be quite dissimilar (i.e., high KL divergence) but still have equal uncertainty (i.e., entropy). For instance, a Beta(10, 1) distribution and a Beta (1,10) distribution have equal entropy but very different KL divergence. See also the equation of information gain with KL divergence on page 3, line 57.

- Although the manuscript was well written in general, there were a number of grammatical and spelling lapses throughout. One prominent one is the title of the manuscript ("How predictability affects habituation to novelty?"). If the title is phrased as a question, this should be "How does predictability affect habituation to novelty"; if the title is not phrased as a question then "How predictability affects habituation to novelty" would be grammatical.

- The derivations of the Bayesian update rule (Equation 6) and the information gain (Equation 8) are non-trivial, but only the results of these derivations (and not the process by which the equations are derived) are presented. Speaking personally, this made it rather difficult for me to check the logic of the manuscript's derivation. I would recommend including additional supplementary material detailing these derivations to aid the interested reader.

Reviewer #2: 1. Is the manuscript technically sound, and do the data support the conclusions?

The manuscript is technically sound. The authors describe a prediction derived from a previously published Bayesian model of novelty and report a single experiment to test that prediction. The results were consistent with the predictions of the model.

2. Has the statistical analysis been performed appropriately and rigorously?

Appropriate statistical analyses are reported, but for ANOVA I would expect to see some indication of effect size estimates and associated confidence intervals.

3. Have the authors made all data underlying the findings in their manuscript fully available?

Summary data are available. I would have liked the authors to give access to the stimulus videos also. It is possible to work out what they showed with reference to the methods and table, but the actual stimuli would help.

4. Is the manuscript presented in an intelligible fashion and written in standard English?

The manuscript is written in standard English, and is generally intelligible, but I think that the very technical nature of the writing will make it difficult to access by all but a specialist audience. With relatively little effort. I am sure that the authors could make the manuscript accessible to a wider readership by simply explaining key terms and concepts in simpler ways and providing more detail.

The concept of long-term novelty (page 1, line 38) might seem counterintuitive or even oxymoronic to some readers.

The authors refer to predictability and uncertainty throughout the introduction of the manuscript, but these terms are first defined at the end of page six, and then in mostly mathematical terms. A clear behavioural/operational definition of predictability and uncertainty when the terms are first used would greatly improve the intelligibility of the manuscript.

Clarity of expression could also be improved in places. The opening paragraph of the introduction, for example, states (lines 36-37) “Therefore, if one is experiencing unpleasant novel events, one should get used to them earlier; if one is experiencing pleasant novel events, one should be as unaccustomed as possible.” I assume that the authors mean that it would be desirable/advantageous for the individual to get used to novel unpleasant events as soon as possible to reduce their impact, but the way this sentence is written, it could be interpreted as meaning that novel unpleasant events lead to rapid habituation.

In the procedure (3.1.3), the authors refer to eight videos, each shown 60 times, but then in the analysis section (3.1.5) mention four stimulus types and 120 exposures. It took me longer than it should have to work out that there were two videos of each stimulus type – a statement to this effect in the procedure would have helped.

6. PLOS authors have the option to publish the peer review history of their article (what does this mean?). If published, this will include your full peer review and any attached files.

Reviewer #1: No

Reviewer #2: No

---

## [Author Response · Author response to Decision Letter 0]

12 Feb 2021

To reviewer 1,

We are grateful to reviewer #1 for the critical comments and useful suggestions that have helped us to improve our paper considerably. As indicated in the response that follow, we have taken the comments and suggestions into account in the revised version of our paper.

1. The empirical section of this manuscript reports a sample of N=8 participants. After exclusion of 1 participant due to excessive eye-blink artefacts, therefore, the final results of this manuscript are derived from a sample of N=7 participants. The effect of this is that the manuscript is severely underpowered to estimate the effect size predicted by its simulations (between initial uncertainty and trial number). This problem is not unique to this manuscript; for instance, Button et al. (Nature Reviews Neuroscience, 2013) discuss the prevalence of underpowered studies in neuroscience more broadly.

There are many consequences of underpowered studies, but the most important one for the present study is that significant results deriving from small samples are likely to significantly overestimate true effect sizes in the population. Notably, this study is underpowered even by the modest standards of ERP research; for instance, Clayson et al. (Psychophysiology, 2019) report the median sample size for ERP studies as 21 participants (3 times that of the sample in this manuscript). Unfortunately, short of replicating results in a larger sample, I believe that this represents a critical shortcoming of the manuscript that precludes interpretation of results.

Response 1:

Thank you for your valuable comment.

According to Yano et al., International Journal of Psychophysiology (2019), the P300 (P3b) experiment of within-participant factors reported that a significant effect could be reliably detected with a sample size of 7. In this study, we conducted a P300 (P3b) experiment with a within-participant factorial design. However, we further increased the number of participants in the experiment to increase the effect detection power. (Clayson et al. (2019) reported on an experiment with a mixed design of between-participant and within-participant factors)

Yano, M., Suwazono, S., Arao, H., Yasunaga, D., & Oishi, H. (2019). Inter-participant variabilities and sample sizes in P300 and P600. International Journal of Psychophysiology, 140, 33-40.

2. The results of the manuscript are presented as a test of a specific computational model of the P300 component of the ERP (in which information gain is operationalised as Kullback-Leibler divergence between posterior and prior beliefs). However, the test of this model presented in the manuscript is qualitative, rather than quantitative:

in both the simulations and the empirical data, there is an interaction between trial number and initial uncertainty. However, such an interaction might also result from other cognitive mechanisms if we assume that their effect is to cause a roughly multiplicative decay in P300 amplitude over time. To give one example, an effect of participant fatigue might also produce the interaction between trial number and initial uncertainty reported in the manuscript.

A far stronger test of the model's predictions would be, rather than discretising trials into 40-trial bins, to run a single-trial regression of P300 amplitudes onto information gain as measured by the manuscript's metric. If this model outperforms one in which P300 amplitude is predicted as a function of trial number (rather than KL divergence), this would constitute stronger evidence for the manuscript's conclusion. This would also be consistent with other recent information-theoretic and Bayesian approaches to analysis of the P300 component (see point 4 below).

Response 2:

Thank you for your valuable comment.

This study aimed to elucidate how predictability affects habituation to novelty using our mathematical model of arousal that we have already tested, subjective evaluation of surprise, and physiological indices (P300) (Yanagisawa et al., Frontiers in Computational Neuroscience, 2019). Therefore, we are not attempting to propose a new model or perform quantitative reverse inference using P300.

However, the reviewer's suggestion was very attractive, so we performed a single-trial regression of the P300 amplitude following the method of Mars et al. (2008). However, the accuracy of estimation of P300 amplitude was worse than that of the conventional averaging method of P300 used in our study, so we could not test our hypothesis. With the P300 amplitude using the conventional averaging method, the analysis of 7 participants in the initial draft and 15 participants in the revised draft showed similar results supporting the hypothesis. Therefore, we thought that we could confirm the correspondence between the model predictions and the physiological indices. However, as pointed out by the reviewer, we cannot deny the effect of fatigue over time, so we stated the following as a limitation of this study.

line 18 in page 15 in “5 Discussion”: “In this study, we experimentally examined habituation associated with repeated presentation of stimuli. Therefore, we cannot deny the possibility that cognitive and physical factors, such as fatigue of the experimental participants over time, may have influenced the interaction between initial uncertainty and initial prediction error. In future experiments, it is necessary to construct an experimental system suitable for single-trial analysis of P300 amplitude, for example, and to examine the time course of habituation in shorter experiments.”

3. An interaction between trial number and initial uncertainty is reported for participants' self-reported surprise ratings. However, I believe that this may be a statistical artefact resulting from the fact that self-report ratings of congruent items are at a floor level (see Figure 5). Indeed, from inspection of the self-report scores in the accompanying supplemental material, it appears that the overwhelming majority of participants reported surprise levels of '1'

(no surprise) for these conditions. In this case it is not surprising that an interaction would occur with initial uncertainty, since surprise can hardly be expected to decrease any further for instruments high in initial certainty. I would also note that in this case, the assumption of the ANOVA (normality of residuals) is violated, and the manuscript might be better served employing a different analytic approach (e.g., probit regression; see Liddell & Kruschke, 2018, Journal of Experimental Social Psychology).

Response 3:

Thank you for your useful suggestion.

Following the suggestion of reviewer 1, we conducted a probit regression analysis, a nonparametric analysis, to test our hypothesis (H1) about the subjective evaluation of surprise. For hypothesis H2, we performed Wilcoxon's signed rank test.

We added the results in line4 page12 in "4 Results".

4. The manuscript's Bayesian model of the amplitude of the P300 is an interesting one, but it does not engage with a body of prior research on Bayesian models of the P300 component of the event-related potential. A Bayesian model of the P300 was first proposed by Kopp (2008; In M. K. Sun (Ed.), Cognitive sciences at the leading edge (pp.

87–96)), and a number of empirical papers testing this and other Bayesian P300 models have been published in the last 5 years (e.g., Kolossa et al., 2013, Frontiers in Human Neuroscience; Kolossa, Kopp & Fingscheidt, 2015, NeuroImage; Bennett et al., 2016, eNeuro; Kopp et al., 2016, Cognitive, Affective, and Behavioural Neuroscience; Bennett et al., 2019, Psychophysiology). The manuscript would be strengthened by engaging with these prior papers that prefigure the ideas proposed here.

Response 4:

Thank you for your useful suggestion.

We cited the paper referred to by the reviewer #1 and added the following sentence in line 21 in page 4.

"Kopp (2007) was the first to study the relationship between Bayesian inference and P300 components (26), and several empirical studies have been conducted since then (27-31). Kolossa et al. showed the correspondence between surprise as the amount of information and P300 (30). They reported that P3a corresponds to KL divergence from Bayesian posterior to prior (termed Bayesian surprise), a change in the probability distribution given a new observation, particularly belief updating about hidden states. On the other hand, P3b is reported to correspond to predictive surprise, meaning surprise about observations under the current probability distribution."

## Minor points

- Page 4, lines 86-87: "Accordingly, the information gain represents a decrease in uncertainty by experiencing an event". This is not quite correct; KL divergence measures the similarity between two probability distributions. These distributions may be quite dissimilar (i.e., high KL divergence) but still have equal uncertainty (i.e., entropy). For instance, a Beta(10, 1) distribution and a Beta (1,10) distribution have equal entropy but very different KL divergence. See also the equation of information gain with KL divergence on page 3, line 57.

Response 5:

We deleted the part you pointed out (following sentence) as it was incorrect.

"Accordingly, the information gain represents a decrease in uncertainty by experiencing an event."

'- Although the manuscript was well written in general, there were a number of grammatical and spelling lapses throughout. One prominent one is the title of the manuscript ("How predictability affects habituation to novelty?"). If the title is phrased as a question, this should be "How does predictability affect habituation to novelty"; if the title is not phrased as a question then "How predictability affects habituation to novelty" would be grammatical.

Response 6:

We corrected the part you pointed out (Title) as follows to make it grammatically correct.

"How predictability affects habituation to novelty"

The derivations of the Bayesian update rule (Equation 6) and the information gain (Equation 8) are non-trivial, but only the results of these derivations (and not the process by which the equations are

derived) are presented. Speaking personally, this made it rather difficult for me to check the logic of the manuscript's derivation. I would recommend including additional supplementary material detailing these derivations to aid the interested reader.

Response 7:

We added the derivation equation as supplementary material.

 

To Reviewer 2

We are grateful to reviewer #2 for the critical comments and useful suggestions that have helped us to improve our paper considerably. As indicated in the response that follow, we have taken the comments and suggestions into account in the revised version of our paper.

Appropriate statistical analyses are reported, but for ANOVA I would expect to see some indication of effect size estimates and associated confidence intervals.

Response 1:

Thank you for your useful suggestion.

We included effect size and confidence intervals for the ANOVA we conducted (in line 16 in page 13).

Summary data are available. I would have liked the authors to give access to the stimulus videos also. It is possible to work out what they showed with reference to the methods and table, but the actual stimuli would help.

Response 2:

Thank you for your useful suggestion.

We added the URLs of the stimulus videos used in this study in line 4 in page 10 in "3.1.2 Stimuli".

The manuscript is written in standard English, and is generally intelligible, but I think that the very technical nature of the writing will make it difficult to access by all but a specialist audience. With relatively little effort. I am sure that the authors could make the manuscript accessible to a wider readership by simply explaining key terms and concepts in simpler ways and providing more detail.

The concept of long-term novelty (page 1, line 38) might seem counterintuitive or even oxymoronic to some readers.

Response 3:

As you pointed out, we revised the text as follows to make it more intuitive and easier to understand.

line 20 in page 2 in "1 Introduction": "to maintain an emotional response to novelty."

The authors refer to predictability and uncertainty throughout the introduction of the manuscript, but these terms are first defined at the end of page six, and then in mostly mathematical terms. A clear behavioural/operational definition of predictability and uncertainty when the terms are first used would greatly improve the intelligibility of the manuscript.

Response 4:

As you pointed out, we added a sentence that defines predictability and uncertainty in behavioral and operational terms.

line 20 in page 3 in "1 Introduction": "Uncertainty is an index for the familiarity of an object, the amount of knowledge an individual has, and so on. Prediction error indicates the difference between the prior prediction and the actual sensation."

Clarity of expression could also be improved in places. The opening paragraph of the introduction, for example, states (lines 36-37) “Therefore, if one is experiencing unpleasant novel events, one should get used to them earlier; if one is experiencing pleasant novel events, one should be as unaccustomed as possible.” I assume that the authors mean that it would be desirable/advantageous for the individual to get used to novel unpleasant events as soon as possible to reduce their impact, but the way this sentence is written, it could be interpreted as meaning that novel unpleasant events lead to rapid habituation.

Response 5: 

In order to avoid misunderstandings due to ambiguity, we corrected the parts you pointed out.

line 18 in page 2 in "1 Introduction": "Therefore, if one is experiencing unpleasant novel events, one should get used to them as soon as possible"

In the procedure (3.1.3), the authors refer to eight videos, each shown 60 times, but then in the analysis section (3.1.5) mention four stimulus types and 120 exposures. It took me longer than it should have to work out that there were two videos of each stimulus type – a statement to this effect in the procedure would have helped.

Response 6:

As you pointed out, it is difficult to understand, so we added the following sentence.

line 7 in page 11 in "3.1.3 Procedure": "Each of the four stimulus types (i.e., AX, AY, BX, and BY) contains two videos (Table 1)."

---

## [Decision Letter · Decision Letter 1]

7 Apr 2021

PONE-D-20-22242R1

How predictability affects habituation to novelty

PLOS ONE

Dear Dr. Yanagisawa,

Thank you for submitting your manuscript to PLOS ONE. After careful consideration, we feel that it has merit but does not fully meet PLOS ONE’s publication criteria as it currently stands. Therefore, we invite you to submit a revised version of the manuscript that addresses the points raised during the review process.

We look forward to receiving your revised manuscript.

Kind regards,

David K Sewell

Academic Editor

PLOS ONE

Journal Requirements:

Additional Editor Comments (if provided):

Dear Dr Yanagisawa,

Thank you for revising your manuscript. You’ll see that the Reviewers are both satisfied with the changes you’ve made to the manuscript, as am I. I am therefore happy to conditionally accept the manuscript, subject to addressing one outstanding issue, detailed below.

Reviewer 1 flags one final point regarding their suggestion for conducting a single-trial regression analysis on the P300 data. I share their impression that noisy individual-level data does not necessarily preclude conducting the regression analysis. Given that you have performed the analysis, I think it would be prudent to report those results, even if they do not lead to definitive conclusions (e.g., if the disaggregated data are too noisy). It would be fine to report this analysis as a footnote to the primary analysis of the binned data. There, you could note any caveats or limitations of the single-trial analysis, also acknowledging its more exploratory nature. Of course, if I am overlooking any obvious reason why the single-trial analysis cannot be run, feel free to clarify this point.

Yours sincerely,

David Sewell

Reviewers' comments:

Reviewer's Responses to Questions

**Comments to the Author**

1. If the authors have adequately addressed your comments raised in a previous round of review and you feel that this manuscript is now acceptable for publication, you may indicate that here to bypass the “Comments to the Author” section, enter your conflict of interest statement in the “Confidential to Editor” section, and submit your "Accept" recommendation.

Reviewer #1: (No Response)

Reviewer #2: All comments have been addressed

2. Is the manuscript technically sound, and do the data support the conclusions?

Reviewer #1: Yes

Reviewer #2: Yes

3. Has the statistical analysis been performed appropriately and rigorously? 

Reviewer #1: Yes

Reviewer #2: Yes

4. Have the authors made all data underlying the findings in their manuscript fully available?

Reviewer #1: Yes

Reviewer #2: Yes

5. Is the manuscript presented in an intelligible fashion and written in standard English?

Reviewer #1: Yes

Reviewer #2: Yes

6. Review Comments to the Author

Reviewer #1: Thank you for the opportunity to review this revision of the manuscript by Ueda and colleagues. I appreciate the efforts that the authors have gone to to address my previous comments, and feel that my concerns have in large part been addressed by the substantial revisions in this new version of the manuscript.

I have one lingering concern regarding the authors' response to my original point #2, in which I noted that splitting the data into 40-trial bins and analysing across bins represents a rather coarse-grained approach to testing model predictions, and suggested that the authors should employ a single-trial regression approach to allow a more fine-grained test of the model. In their response, the authors report that they did indeed conduct such an analysis, but that they do not report the results of the analysis because "the accuracy of estimation of P300 was worse than that of the conventional averaging method of P300 used in our study, so we could not test our hypothesis".

I am slightly troubled by this response, which seems to side-step the substance of my recommendation. It is natural that single-trial measures of the P300 should have greater variance than the average of 40-trial bins, but I am afraid I do not quite understand why this variance should preclude running a regression analysis.

My inclination is to request further details on this analysis from the authors, including a more detailed account of what features of the data prevented a single-trial regression analysis from being interpretable. However, I should state that I do not feel that this single issue in and of itself would make the difference between whether the manuscript should be published or not. I feel that the results are sufficiently compelling and rigorous in their current form, and would simply be interested to know more about the results of this regression analysis. I do not feel that this issue in and of itself is so major that it would preclude publication. In that light, I defer to the editor's judgment regarding this point.

Reviewer #2: (No Response)

7. PLOS authors have the option to publish the peer review history of their article (what does this mean?). If published, this will include your full peer review and any attached files.

Reviewer #1: No

Reviewer #2: No

---

## [Author Response · Author response to Decision Letter 1]

14 May 2021

Dear Editor,

We are most grateful to you and the reviewer for the helpful comments on the revised version of our manuscript. In response to the reviewers' comments, we have revised the manuscript.

We hope that the revised version of our paper is now suitable for publication in PLOS ONE and we look forward to hearing from you at your earliest convenience.

Sincerely,

Kazutaka Ueda, Ph.D. 

Hideyoshi Yanagisawa, Ph.D.

To Editor,

(Editor’s comment) Reviewer 1 flags one final point regarding their suggestion for conducting a single-trial regression analysis on the P300 data. I share their impression that noisy individual-level data does not necessarily preclude conducting the regression analysis. Given that you have performed the analysis, I think it would be prudent to report those results, even if they do not lead to definitive conclusions (e.g., if the disaggregated data are too noisy). It would be fine to report this analysis as a footnote to the primary analysis of the binned data. There, you could note any caveats or limitations of the single-trial analysis, also acknowledging its more exploratory nature.

Response : Thank you very much for your valuable comments.

Following the suggestion of reviewer 1, we estimated the P300 amplitude for each trial according to the methods of a previous study (Mars et al., 2008). We performed a single-trial regression for each participant using estimates of the P300 amplitude for each condition; however, not all of the regressions were significant (eight significant regressions out of 60 analyses). We believe that we had only a single P300 amplitude for each time point for each participant in the experiment, and the background EEG and artifacts affected each trial randomly, thereby preventing an accurate regression. We described these in the supplementary material section and shared a scatter plot for each condition of the time transition of the P300 amplitude estimated in each trial for each participant as supplementary material.

We added the following sentence in supplementary material section.

“In order to examine the detailed time trends of the P300 amplitude related to habituation, we estimated the P300 amplitude for each trial according to the methods of a previous study (Mars et al., 2008). We performed a single-trial regression for each participant using estimates of the P300 amplitude for each condition; however, not all of the regressions were significant (eight significant regressions out of 60 analyses). This may be because only a single P300 amplitude was obtained for each time point for each participant in the experiment, and the background EEG as well as artifacts affected each trial randomly, thereby preventing an accurate regression. Therefore, in this study, the averaging method, which is commonly used in P300 studies, was used to analyze the data with a higher signal-to-noise ratio by reducing the effects of background EEG and artifacts. For reference, a scatter plot for each condition of the time transition of the P300 amplitude estimated in each trial for each participant is shared as supplementary material.”

To reviewer 1,

We are grateful to reviewer #1 for the critical comments and useful suggestions that have helped us to improve our paper considerably. As indicated in the response that follow, we have taken the comments and suggestions into account in the revised version of our paper.

Response:

Thank you very much for your valuable comments.

Following your suggestion, we estimated the P300 amplitude for each trial according to the methods of a previous study (Mars et al., 2008), and performed a single-trial regression for each participant using estimates of the P300 amplitude for each condition; however, not all of the regressions were significant (eight significant regressions out of 60 analyses). We believe that we had only a single P300 amplitude for each time point for each participant in the experiment, and the background EEG and artifacts affected each trial randomly, thereby preventing an accurate regression. We described these in the supplementary material section and shared a scatter plot for each condition of the time transition of the P300 amplitude estimated in each trial for each participant as supplementary material.

We added the following sentence in supplementary material.

“In order to examine the detailed time trends of the P300 amplitude related to habituation, we estimated the P300 amplitude for each trial according to the methods of a previous study (Mars et al., 2008). We performed a single-trial regression for each participant using estimates of the P300 amplitude for each condition; however, not all of the regressions were significant (eight significant regressions out of 60 analyses). This may be because only a single P300 amplitude was obtained for each time point for each participant in the experiment, and the background EEG as well as artifacts affected each trial randomly, thereby preventing an accurate regression. Therefore, in this study, the averaging method, which is commonly used in P300 studies, was used to analyze the data with a higher signal-to-noise ratio by reducing the effects of background EEG and artifacts. For reference, a scatter plot for each condition of the time transition of the P300 amplitude estimated in each trial for each participant is shared as supplementary material.”

---

## [Editor Report · Decision Letter 2]

17 May 2021

How predictability affects habituation to novelty

PONE-D-20-22242R2

Dear Dr. Yanagisawa,

We’re pleased to inform you that your manuscript has been judged scientifically suitable for publication and will be formally accepted for publication once it meets all outstanding technical requirements.

Kind regards,

David K Sewell

Academic Editor

PLOS ONE

Additional Editor Comments (optional):

Dear Dr Yanagisawa,

Many thanks for your detailed responses to the comments and requests of the reviewers. I think the changes that you've made to the manuscript have greatly improved it, and am delighted to accept your manuscript for publication in PLOS ONE.

Yours sincerely,

David K Sewell
---

## [Editor Report · Acceptance letter]

19 May 2021

PONE-D-20-22242R2 

How predictability affects habituation to novelty 

Dear Dr. Yanagisawa:

I'm pleased to inform you that your manuscript has been deemed suitable for publication in PLOS ONE. Congratulations! Your manuscript is now with our production department. 

Kind regards, 

on behalf of

Dr. David Keisuke Sewell 

Academic Editor

PLOS ONE